# Arsenic exposure is associated with elevated sweat chloride concentration and airflow obstruction among adults in Bangladesh: A cross-sectional study

Mi-Sun S. Lee[1],[⊛], Crystal M. North[1,2],[⊛], Irada Choudhuri[3], Subrata K. Biswas[4], Abby F. Fleisch[5,6], Afifah Farooque[7], Diane Bao[7], Sakila Afroz[8], Sadia Mow[8], Nazmul Husain[8], Fuadul Islam[8], Md Golam Mostofa[8], Partha Pratim Biswas[9], David S. Ludwig[10], Subba R. Digumarthy[11], Christopher Hug[12], Quazi Quamruzzaman[8], David C. Christiani[1,2], Maitreyi Mazumdar[1,7]*

1 Department of Environmental Health, Harvard T.H. Chan School of Public Health, Boston, Massachusetts, United States of America, 2 Division of Pulmonary and Critical Care Medicine, Massachusetts General Hospital, Boston, Massachusetts, United States of America, 3 Department of Medicine, University of Pittsburgh Medical Center, Pittsburgh, Pennsylvania, United States of America, 4 Department of Molecular and Cell Biology, University of Connecticut, Storrs, Connecticut, United States of America, 5 Center for Interdisciplinary Population Health Research, MaineHealth, Portland, Maine, United States of America, 6 Pediatric Endocrinology and Diabetes, Maine Medical Center, Portland, Maine, United States of America, 7 Department of Neurology, Boston Children's Hospital, Boston, Massachusetts, United States of America, 8 Dhaka Community Hospital Trust, Dhaka Bangladesh, 9 Department of Biochemistry, Bangabandhu Sheikh Mujib Medical University, Dhaka, Bangladesh, 10 New Balance Obesity Prevention Center, Boston Children's Hospital, Boston, Massachusetts, United States of America, 11 Thoracic Imaging and Intervention Division, Massachusetts General Hospital, Boston, Massachusetts, United States of America, 12 Consultant, Brookline, Massachusetts, United States of America,

⊛ These authors contributed equally to this work.
* maitreyi.mazumdar@childrens.harvard.edu

## Abstract

Arsenic is associated with lung disease and experimental models suggest that arsenic-induced degradation of the chloride channel CFTR (cystic fibrosis transmembrane conductance regulator) is a mechanism of arsenic toxicity. We examined associations between arsenic exposure, sweat chloride concentration (measure of CFTR function), and pulmonary function among 269 adults in Bangladesh. Participants with sweat chloride ≥ 60 mmol/L had higher arsenic exposures than those with sweat chloride < 60 mmol/L (water: median 77.5 µg/L versus 34.0 µg/L, $p = 0.025$; toenails: median 4.8 µg/g versus 3.7 µg/g, $p = 0.024$). In linear regression models, a one-unit µg/g increment in toenail arsenic was associated with a 0.59 mmol/L higher sweat chloride concentration, $p < 0.001$. Among the entire study population, after adjusting for covariates including age, sex, smoking, education, and height, toenail arsenic concentration was associated with increased odds of airway obstruction (OR: 1.97, 95%: 1.06, 3.67, $p = 0.03$); however, sweat chloride concentration did not mediate this association. Our findings suggest that sweat chloride concentration may serve

**Data availability statement:** We have uploaded a minimal anonymized dataset to the Harvard Dataverse. The dataset can cam be found at the following location: https://dataverse.harvard.edu/dataset.xhtml?persistentId=doi:10.7910/DVN/OLY5P2

**Funding:** We received funding from the National Institute of Environmental Health Sciences (www.niehs.nih.gov) ((NIEHS-R01-ES027825) M.M., (NIEHS-R01-ES011622) D.C.C., (NIEHS-P30-ES000002) M.M.) and from the National Heart, Lung, and Blood Institute (www.nhlbi.nih.gov) (NHLBI-K23-HL154863) C.M.N. The contents are solely the responsibility of the authors and do not represent the official views of the National Institutes of Health.

**Competing interests:** The authors have declared that no competing interests exist.

as novel biomarker for arsenic exposure, warranting further investigation in diverse populations, and that arsenic likely acts on the lung through mechanisms other than inducing CFTR dysfunction. Alternative mechanisms by which environmental arsenic exposure may lead to obstructive lung disease, such as arsenic-induced direct lung injury and/or increase lung proteinase activity, require additional exploration in future work.

## Introduction

Up to 220 million people in as many as 70 countries are exposed to elevated arsenic levels through contaminated groundwater [1,2]. Arsenic exposure has been associated with lung dysfunction [3–8], respiratory symptoms [9–11], bronchiectasis [12], and increased tuberculosis-related [13] and respiratory-related mortality [14]. One proposed mechanism for arsenic's effects on the lung is impairment of mucociliary clearance.

The cystic fibrosis transmembrane conductance regulator (CFTR) is a cyclic adenosine monophosphate-regulated chloride channel in the apical membrane of airway epithelial cells. CFTR plays an essential role in mucociliary clearance by establishing an osmotic gradient across the airway epithelium that promotes fluid secretion [15–17]. Mutations in *CFTR*, the gene that encodes CFTR, cause cystic fibrosis (CF), a rare autosomal recessive disease that affects the lungs and digestive system. Cystic fibrosis is also associated with a diminished immune response to some gram-negative bacteria; a similarly diminished immune response is induced by arsenic exposure [18,19].

Recent studies of CFTR in killifish, an environmental model organism, demonstrate that arsenic induces CFTR ubiquitinylation and degradation [20,21]. Subsequent studies translate the killifish model into humans and use human airway epithelial cell culture to demonstrate that arsenic induces an increase in multiubiquitinlyated CFTR, which results in CFTR degradation and reduced CFTR-mediated chloride secretion [22]. Together, these reports demonstrate that arsenic impairs CFTR function and support the hypothesis that there is a shared mechanism driving both cystic fibrosis and arsenic-induced lung disease. Our study aims to further translate these findings to humans and test the hypothesis that arsenic induces CFTR degradation in humans who are exposed to arsenic through the environment.

In humans, CFTR function is directly measured by a sweat test, which is the gold standard diagnostic test for cystic fibrosis [23]. CFTR mediates chloride resorption from sweat; in the absence or impaired function of CFTR, sweat chloride concentration is elevated. In a recent study of 100 adults in Bangladesh—a country with among the highest arsenic exposure in the world—water and toenail arsenic concentrations were higher among adults with increased sweat conductivity (a marker for elevated sweat chloride concentrations), and those with the highest sweat conductivity did not have *CFTR* mutations consistent with a genetic diagnosis of cystic fibrosis [24]. This finding in humans supports the hypothesis that arsenic induces CFTR degradation, as was seen in experimental models, and that CFTR degradation may be a mechanism of arsenic toxicity.

Although sweat conductivity is correlated with sweat chloride concentration, the latter remains the gold standard for identifying CFTR dysfunction [25]. Therefore, to evaluate the relationship between arsenic exposure, CFTR dysfunction, and lung function, we measured arsenic levels, sweat chloride concentrations, and lung function among adults in Bangladesh. Our study aimed to test the specific hypothesis that higher arsenic exposures lead to degradation of the CFTR protein in cell membranes; evidence of CFTR degradation is elevated sweat chloride. Furthermore, we hypothesized that CFTR dysfunction mediates the relationship between arsenic exposure and lung dysfunction.

## Methods

### Ethics statement

All protocols were reviewed and approved by the institutional review boards (IRBs) of Boston Children's Hospital (BCH) (Protocol number: IRB-P00024501), the Dhaka Community Hospital Trust (DCHT), and the Bangabandhu Sheikh Mujib Medical University (BSMMU) (Protocol number: BSMMU/2019/3739). BSMMU received de-identified samples only and did not receive participant information. The Harvard T.H. Chan School of Public Health (HSPH) relied on BCH IRB review. All participants provided written informed consent.

### Inclusivity in global research

Additional information regarding the ethical, cultural, and scientific considerations specific to inclusivity in global research is included in the Supporting Information (S1 Checklist).

### Study participants

From 13/05/2018 through 15/11/2022, we enrolled 300 adults at Dhaka Community Hospital. These participants were recruited from a total of 1800 adults who had participated in studies of arsenic and skin lesions between 2001 and 2003 in Pabna, Bangladesh [26]. We paused recruitment and study visits for one week between 20/12/2020 and 27/12/2020 while renewal of IRB applications was under review. No participants were enrolled from BSMMU. BSMMU laboratories were used to evaluate de-identified samples. We began by creating a list of potential participants using random digit assignment and contacted each potential participant in order on the list until we had reached our target enrollment. All participants from the 2001–2003 study were initially considered eligible for recruitment. Exclusion criteria included evidence of lung cancer or active pulmonary tuberculosis on chest radiography.

### Arsenic exposure measurements

We collected drinking water samples from each participant's home. Water arsenic concentrations were measured in the Environmental Engineering Laboratory at the Bangladesh University of Engineering and Technology using graphite furnace atomic absorption spectrometry (GF-AAS) [27]. The limit of detection (LOD) for arsenic was 1 µg/L. Drinking water arsenic concentrations from the 2001–2003 study were measured using inductively coupled plasma mass spectrometry (ICP-MS) by the Environmental Laboratory Services (North Syracuse, NY, USA) [26].

We collected toenail clippings from all ten toes on a white sheet of paper, placed them in a small coin envelope (ULINE Model No. S-7798) and stored them at room temperature. The LOD for arsenic was 0.002 µg/g. The Dartmouth Trace Element core facility measured arsenic in toenails using ICP-MS [28]. Arsenic concentrations in toenail clippings from the 2001–2003 study were measured at the Harvard T.H. Chan School of Public Health using similar ICP-MS protocols [29].

### Sweat chloride measurements

We performed sweat collection and sweat chloride testing according to established guidelines [30]. Briefly, we obtained sweat from the ventral surface of both forearms of each participant. After cleaning the skin with distilled water, we stimulated sweat secretion by iontophoresis (total applied current 1.5 mA, 50 µA/cm²) for five minutes using 0.5% pilocarpine

gel disks (Pilogels®) and sweat was collected for 30 minutes in a Macroduct® Sweat Collection System (Wescor Biomedical Systems, Logan, UT). The laboratory measured sweat chloride concentrations within 48 hours at the Department of Biochemistry, Bangabandhu Sheikh Mujib Medical University (BSMMU) using a Dimension® RxL Max® Clinical Chemistry Analyzer (Siemens Healthcare Diagnostics Inc., Newark, DE) with QuikLYTE® Integrated Multisensor Technology (IMT). The LOD and coefficient of variation (CV) were 10 mmol/L and 1.1%, respectively. Quality control measures included not analyzing samples with sweat volumes less than 15 µL and including sweat controls with standardized chloride concentrations in each analytical batch.

### DNA isolation and *CFTR* sequence analysis

For participants with sweat chloride concentrations ≥ 60 mmol/L, we retrieved archived DNA samples that were obtained in the 2001–2003 study. DNA had been extracted from whole blood samples using the Puregene DNA Isolation kit (Gentra Systems, Minneapolis, MN) and then stored at -80°C. We sent retrieved DNA samples to Ambry Genetics® (Ambry Test: CF; Aliso Viejo, CA) for *CFTR* sequence analysis and deletion/duplication analysis [31].

### Pulmonary function testing (PFT) and questionnaires

We measured lung function using EasyOne® Air handheld spirometers (Medical Technologies, Andover, MA). All participants had abstained from smoking for at least 45 minutes prior to testing and completed testing in the seated position after loosening any tight-fitting clothing. Spirometry was conducted in accordance with American Thoracic Society (ATS) guidelines by trained study staff, and participants were given up to seven tries to achieve three ATS acceptable and reproducible maneuvers [32]. Measured parameters included forced expiratory volume in one second ($FEV_1$) and forced vital capacity (FVC), and all participants repeated pulmonary function tests (PFTs) after receiving four puffs of salbutamol (Beximco Pharmaceuticals, Bangladesh) and waiting for 10 minutes. Study staff monitored data quality control per ATS guidelines by assessing the accuracy of spirometer measurements with volume and multi-flow calibration using a 3-litre syringe (Medical Technologies, Andover, MA) each day before study procedures began to ensure that measured flow and volumes remained within ± 3.5%, per ATS guidelines [32]. We stopped performing PFTs in March 2020, midway through participant enrollment, for the safety of participants and study staff because of the COVID-19 pandemic.

Two study investigators manually reviewed all PFTs for ATS acceptability and reproducibility standards using ATS guidelines for spirometry standardization and for interpretation of lung function tests [32,33]. We included PFTs in the analytic cohort if there were at least two acceptable maneuvers and both the $FEV_1$ and FVC from the two best trials were within 200mL of one another. Per the Global Initiative for Chronic Obstructive Lung Disease (GOLD), we defined airflow obstruction as a post-bronchodilator $FEV_1/FVC < 0.7$ [34].

We used structured questionnaires to collect information on age, education, occupation, smoking history, and medications. We characterized respiratory symptoms using the American Thoracic Society's Division of Lung Disease questionnaire (ATS-DLD 1978) [35]. We measured height and weight at the study visit.

### Chest radiographs

All participants underwent posteroanterior and lateral chest radiographs performed by a trained radiology technician at Dhaka Community Hospital (DCH). All radiographs were evaluated by a DCH staff radiologist who was unaware of arsenic exposure, sweat chloride, or PFT results. The staff radiologist evaluated the radiographs for the presence of parenchymal, bony, or cardiac abnormalities. For participants who were diagnosed with airflow obstruction, a senior radiologist provided additional imaging review using a structured data collection form.

## Statistical analysis

We first examined all variables using descriptive statistics and assessed potential differences in characteristics between groups using *t*-tests for continuous variables and Chi-square tests for categorical variables. We computed Spearman correlation coefficients (*r*) between water and toenail arsenic concentrations. For samples below the LOD, we estimated arsenic concentrations as LOD/2. We used the average of the toenail arsenic concentrations from the 2001–2003 and 2018–2022 visits to estimate long-term arsenic exposure. Our primary exposure of interest was toenail arsenic concentration. We modeled toenail arsenic using untransformed values and interquartile ranges (IQR).

We used multivariable linear regression to characterize the relationship between arsenic and sweat chloride. We evaluated for potential confounding effects by age, sex, smoking status (current, former, and never), and education (able to write, primary education, middle school or above) by placing each variable in the models individually and assessing for change in the regression coefficient for the arsenic measure.

We also used logistic regression to estimate the relationship between toenail arsenic concentration and the odds of having a sweat chloride concentration ≥ 60 mmol/L. We chose this threshold to be consistent with diagnostic criteria for cystic fibrosis. We considered toenail arsenic concentration our primary measure of arsenic exposure. In sensitivity analyses, we tested the above-described relationships using lower sweat chloride thresholds (≥ 50 mmol/L, ≥ 40 mmol/L, and ≥ 30 mmol/L), as these lower thresholds are sometimes used to identify those who need additional evaluation for cystic fibrosis [25].

We then characterized the relationships between toenail arsenic concentration and lung function using multivariable linear regression ($FEV_1$, FVC, $FEV_1$/FVC) or logistic regression ($FEV_1$/FVC < 0.7), adjusting for age, sex, smoking status, education, and height (in centimeters). Lastly, we employed mediation analyses to estimate the degree to which sweat chloride mediates the relationship between long-term arsenic exposure and lung function, adjusting for age, sex, smoking status, education, and height. Results are presented as regression coefficients (β with 95% confidence interval [95% CI]) for continuous outcomes and odds ratio (OR with 95% CI) for dichotomous outcomes per IQR increment in toenail arsenic concentrations. In sensitivity analyses, we restricted analyses to never smokers to rule out the potential masking effects of smoking on the relationship between arsenic and lung function.

We performed statistical analyses using SAS (version 9.4; SAS Institute, Inc.), R (version 4.2.2; R Development Core Team), and RStudio (version 2022.12.0 + 353) software.

## Results

### Study population

Of the 300 enrolled participants, 31 participants were excluded due to missing toenail clippings (n = 10) or insufficient sweat volume (n = 21), for a total of 269 (90%) participants. Participant characteristics are shown in Table 1. Of the 300 enrolled participants 166 (55.3%) completed PFTs that met ATS acceptability and reproducibility criteria prior to the COVID19-related shutdown. The 166 participants who underwent PFTs were more frequently women (53% vs 33.6% p < 0.001) and generally less educated (p = 0.02) than the 134 participants for whom PFTs were not performed; otherwise there were no significant differences between these groups (S1 Table).

### Arsenic concentrations

The distributions of drinking water and toenail arsenic concentrations are shown in Table 2. Overall, drinking water and toenail arsenic concentrations in 2018–2022 were lower than those observed in 2001–2003 (water median 41.0 µg/L in 2018–2021 compared with 71.2 µg/L in 2001–2003), reflecting the impact of arsenic awareness and reduction activities. We observed no significant sex differences in drinking water or toenail arsenic concentrations. All toenail arsenic concentrations were above the LOD of 0.002 µg/g. Comprehensive visual inspection of toenail arsenic concentrations using

**Table 1. Characteristics of study population.**

| Variables | All Participants (*n*=269) | | | Participants with PFTs (*n*=166) | | |
|---|---|---|---|---|---|---|
| | All (*n*=269) | Men (*n*=147) | Women (*n*=122) | All (*n*=166) | Men (*n*=78) | Women (*n*=88) |
| Age (y) | 51.4±10.4 | 50.8±11.6 | 52.2±8.9 | 51.3±9.9 | 50.9±11.0 | 51.6±8.8 |
| BMI (kg/m²) | 22.8±3.7 | 22.6±3.6 | 23.1±3.8 | 22.7±3.8 | 22.0±3.6 | 23.2±3.9 |
| Education | | | | | | |
| Able to write | 105 (39.0) | 39 (26.5) | 66 (54.1) | 71 (42.8) | 22 (28.2) | 49 (55.7) |
| Primary education | 68 (25.3) | 37 (25.2) | 31 (25.4) | 45 (27.1) | 27 (34.6) | 18 (20.5) |
| Middle school or above | 96 (35.7) | 71 (48.3) | 25 (20.5) | 50 (30.1) | 29 (37.2) | 21 (23.9) |
| Occupation | | | | | | |
| Agriculture | 65 (24.2) | 65 (44.2) | – | 31 (18.7) | 31 (39.7) | – |
| Business | 46 (17.1) | 46 (31.3) | – | 28 (16.9) | 28 (35.9) | – |
| Service | 27 (10.0) | 25 (17.0) | 2 (1.6) | 11 (6.6) | 10 (12.8) | 1 (1.1) |
| Laborer | 11 (4.1) | 9 (6.1) | 2 (1.6) | 10 (6.0) | 8 (10.3) | 2 (2.3) |
| Retired | 2 (0.7) | 2 (1.4) | – | 1 (0.6) | 1 (1.3) | – |
| Housewife | 118 (43.9) | – | 118 (96.7) | 85 (51.2) | – | 85 (96.6) |
| Smoking status | | | | | | |
| Current smoker | 49 (18.2) | 49 (33.3) | – | 31 (18.7) | 31 (39.7) | – |
| Former | 32 (11.9) | 32 (21.8) | – | 20 (12.1) | 20 (25.6) | – |
| Never | 188 (69.9) | 66 (44.9) | 122 (100.0) | 115 (69.3) | 27 (34.6) | 88 (100.0) |
| Respiratory symptoms | | | | | | |
| Cough | 66 (24.5) | 40 (27.2) | 26 (21.3) | 47 (28.3) | 23 (29.5) | 24 (27.3) |
| Shortness of breath | 37 (13.8) | 20 (13.6) | 17 (13.9) | 27 (16.3) | 14 (17.9) | 13 (14.8) |
| Any cough or shortness of breath | 77 (28.6) | 45 (30.6) | 32 (26.2) | 54 (32.5) | 26 (33.3) | 28 (31.8) |

All results are presented as n(%), or mean±standard deviation (SD).

BMI, body mass index.

histograms and kernel density plots, along with residual distribution analysis (data not shown) suggested that the data appeared approximately normally distributed. Toenail arsenic concentrations were moderately correlated with drinking water arsenic concentrations ($r=0.66$, $p<0.0001$).

## Sweat chloride concentrations

Mean±standard deviation (SD) sweat chloride concentration was 49.1±16.8 (median: 48.0 mmol/L, range: 18.0 to 98.0 mmol/L; IQR: 24.0 mmol/L). A total of 69 (25.7%) participants had sweat chloride concentration≥60 mmol/L. Sweat chloride concentrations were higher in men than women (51.6±16.8 vs. 46.1±16.4, $p=0.007$) and higher among smokers vs. never smokers (52.6±16.5 vs. 47.6±16.8). There were no significant associations between sweat chloride concentration, participant's age and education (S2 Table).

## *CFTR* sequence analysis

Of the 69 participants with sweat chloride≥60 mmol/L, 45 (65.2%) underwent *CFTR* sequence analysis. None of the DNA samples tested revealed homozygous pathogenic mutations in *CFTR*. Six participants had heterozygous findings consistent with classification as cystic fibrosis carriers (S3 Table). These findings suggest that the elevated sweat chloride concentrations we observed were not caused by biallelic *CFTR* mutations that lead to absent or abnormal CFTR protein production.

**Table 2. Distribution of arsenic concentrations in drinking water and toenails.**

| Arsenic concentration | Mean±SD | Min | 25th percentile | Median | 75th percentile | Max | IQR |
|---|---|---|---|---|---|---|---|
| **All subjects (*n*=269)** | | | | | | | |
| Drinking water, μg/L | | | | | | | |
| 2001-2003 | 234.3±300.4 | <LOD | 16.8 | 71.2 | 410.5 | 1190.0 | 393.7 |
| 2018-2021 | 138.5±196.3 | <LOD | 9.5 | 41.0 | 202.0 | 1239.0 | 192.5 |
| Toenails, μg/g | | | | | | | |
| 2001-2003 | 6.9±6.7 | 0.3 | 2.3 | 4.8 | 9.4 | 45.9 | 7.1 |
| 2018-2021 | 5.8±5.8 | 0.2 | 1.4 | 4.1 | 8.3 | 31.6 | 6.9 |
| Long-term† | 6.3±5.4 | 0.3 | 2.2 | 4.7 | 8.8 | 29.1 | 6.6 |
| **Subjects with PFTs (*n*=166)** | | | | | | | |
| Drinking water, μg/L | | | | | | | |
| 2001-2003 | 293.3±328.7 | <LOD | 33.6 | 99.0 | 484.5 | 1190.0 | 450.9 |
| 2018-2021 | 166.9±206.8 | <LOD | 24.0 | 82.5 | 247.0 | 1239.0 | 223.0 |
| Toenails, μg/g | | | | | | | |
| 2001-2003 | 8.2±6.8 | 0.3 | 3.7 | 6.9 | 10.0 | 45.9 | 6.3 |
| 2018-2021 | 7.0±6.3 | 0.3 | 2.3 | 5.4 | 9.8 | 31.6 | 7.5 |
| Long-term† | 7.6±5.3 | 0.4 | 3.7 | 6.9 | 10.4 | 29.1 | 6.7 |

†Average of toenail arsenic concentrations in 2001–2003 and 2018–2021.

IQR, interquartile range; SD, standard deviation.

LOD, limit of detection.

## Pulmonary function tests, respiratory symptoms, and chest imaging

No participant was found to have lung cancer, active pulmonary tuberculosis, cardiac or bony abnormalities on chest films. Of the 166 who underwent PFTs, mean±SD was 1.97±0.54 L for $FEV_1$, 2.57±0.67 for FVC, and 0.77±0.09 for $FEV_1$/FVC (Table 3). Airflow obstruction was present among 19.9% (n=33) of the study cohort. Most of the men with airflow obstruction (n=23, 85%) had a history of smoking, while none of the women with airflow obstruction had a history of smoking (p<0.001). A total of 55 participants (33.1%) endorsed respiratory symptoms, which included wheezing, shortness of breath or coughing. Respiratory symptoms were more prevalent among those with airflow obstruction; 57.6% of people with airflow obstruction reported respiratory symptoms compared to 27.1% of people without airflow obstruction (data not shown). Among the 33 participants with airflow obstruction identified on PFTs, additional review of chest imaging showed evidence of bronchiectasis in 5 individuals.

## Arsenic and sweat chloride

To evaluate sweat chloride's potential use as a biomarker of arsenic exposure, we present the most parsimonious models including modeling arsenic on a natural scale for ease of interpretation. Our decisions to use untransformed arsenic values and parametric models are supported by a comprehensive visual inspection using kernel density plots, along with residual distribution analysis (data not shown). These analyses suggested that our data appeared normally distributed and the adequacy of our study's sample size [36,37]. In our primary analyses, we found that concurrent drinking water and toenail arsenic concentrations were higher among participants with sweat chloride ≥60 mmol/L compared to <60 mmol/L (water: median 77.5 μg/L versus 34.0 μg/L, p=0.025; toenails: median 4.8 μg/g versus 3.7 μg/g, p=0.024) (Fig 1). A similar pattern was seen for long-term arsenic concentration (S1 Fig). In models adjusted for sex and smoking status, a one-unit μg/g increment in toenail arsenic concentration was associated with a 0.59 mmol/L higher sweat chloride concentration, p<0.001, and a one-unit μg/g increment in toenail arsenic concentration was associated with greater odds

**Table 3. Pulmonary function tests after bronchodilator treatment (*n* = 166).**

| Measures | All | Men | Women |
|---|---|---|---|
| FEV$_1$, L | 1.97 ± 0.54 | 2.27 ± 0.58 | 1.70 ± 0.33 |
| FVC, L | 2.57 ± 0.67 | 3.05 ± 0.61 | 2.14 ± 0.35 |
| FEV$_1$/FVC | 0.77 ± 0.09 | 0.74 ± 0.11 | 0.80 ± 0.06 |
| Airflow obstruction[†] | 33 (19.9) | 27 (34.6) | 6 (6.8) |

*n* (%) or Mean ± SD.

[†]Any participant whose FEV$_1$/FVC < 0.7 was considered an individual with airflow obstruction.

SD, standard deviation

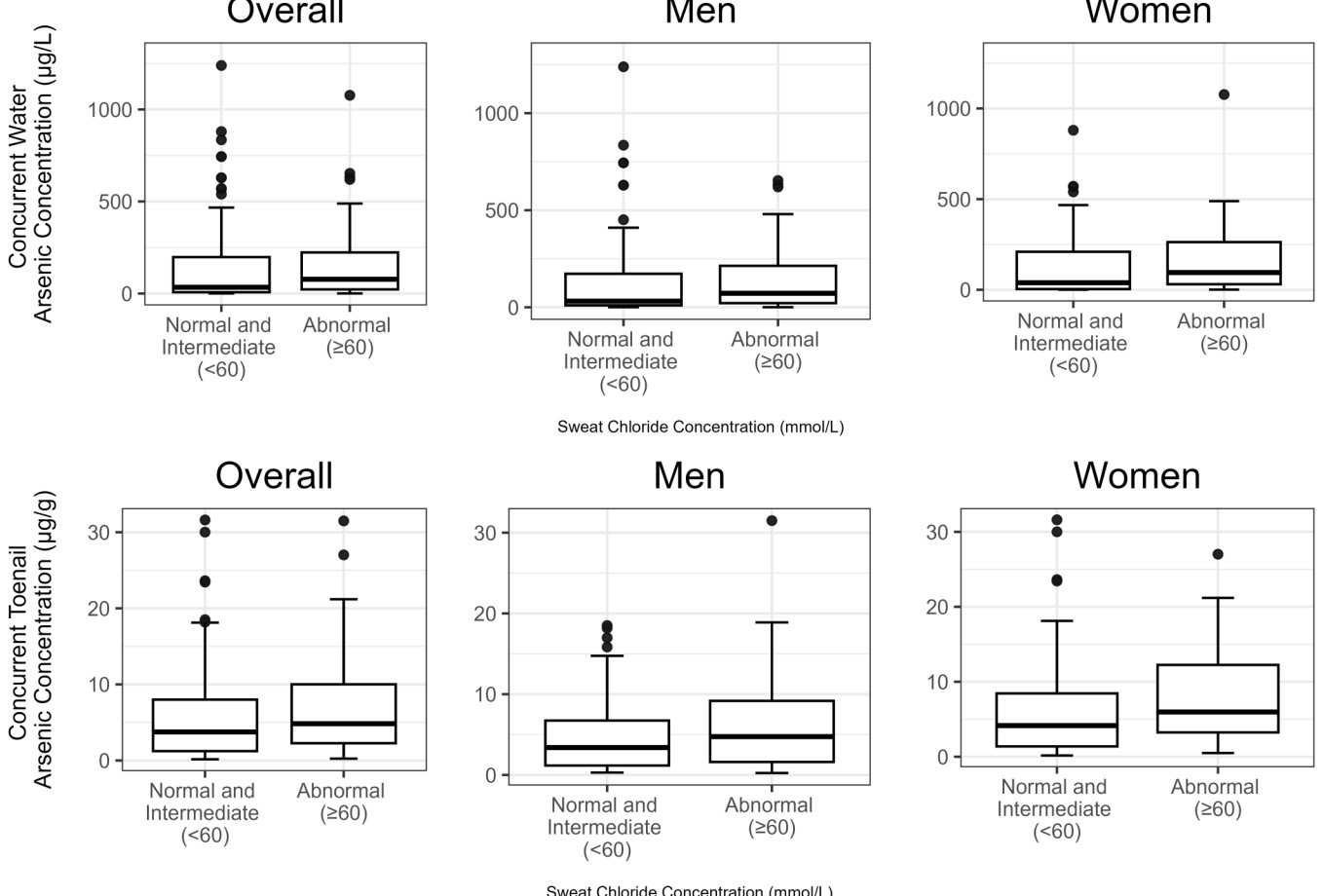

**Fig 1. Arsenic concentrations in drinking water and toenails by sweat chloride concentration.** Distribution of concurrent drinking water and toenail arsenic concentrations among participants with normal/intermediate (<60 mmol/L) and abnormal (≥60 mmol/L) sweat chloride concentrations.

of having a sweat chloride concentration ≥ 60 mmol/L (OR: 1.06, 95% CI, 1.01, 1.11). We found similar results at lower thresholds of sweat chloride. Models examining varying sweat chloride cutoffs are shown in S2–3 Figs. In models fully adjusted for covariates, an IQR (6.6 µg/g) increment in long-term exposure to toenail arsenic was associated with a 3.2 mmol/L (95% CI: 0.57, 5.85, *p* = 0.017) higher sweat chloride concentration (S4 Fig).

## Arsenic and pulmonary function tests

In adjusted models, an IQR increment in toenail arsenic concentration at each timepoint was associated with nonsignificant reductions in $FEV_1$, FVC, and $FEV_1$/FVC (Table 4). An IQR increment in toenail arsenic concentration from 2001–2003 was associated with greater odds of airflow obstruction (toenail arsenic 2001–2003 OR: 1.57, 95% CI: 1.03, 2.40, $p = 0.03$; long-term toenail arsenic OR: 1.97, 95%: 1.06, 3.67, $p = 0.03$). When restricting the analytic cohort to the 115 never smokers, relationships between arsenic exposure and the odds of airflow obstruction were even more pronounced (toenail arsenic 2001–2003 OR: 1.97, 95% CI 1.13, 3.45, $p = 0.02$; toenail arsenic 2018–2021 OR: 2.40, 95% CI 1.02, 5.65, $p = 0.05$; long-term toenail arsenic OR: 3.97, 95% CI 1.44, 10.90, $p = 0.008$; S4 Table).

## Mediation analysis

Mediation analyses indicate that there is no evidence of sweat chloride as a mediator of the relationship between toenail arsenic concentrations and airflow obstruction (S4 Fig).

## Discussion

Our study demonstrates that 1) elevated sweat chloride concentration, the diagnostic hallmark of cystic fibrosis, is found among individuals exposed to arsenic through drinking water, 2) individuals with sweat chloride concentrations ≥ 60 mmol/L did not have a genetic diagnosis of cystic fibrosis, 3) long-term arsenic exposure as estimated by toenail arsenic concentration is associated with airflow obstruction, particularly among those who have never smoked, and 4) sweat chloride concentration did not mediate the association between arsenic and airflow obstruction. These findings suggest that high sweat chloride may be a novel biomarker of arsenic exposure, but that CFTR dysfunction is unlikely to be the primary mechanism of arsenic-associated lung disease.

We used toenail arsenic concentrations to estimate internal arsenic dose. Toenail arsenic is considered good biomarker for long-term exposure when external exposures are consistent [38]. In secondary analyses, despite the modest correlation between toenail and drinking water arsenic, we found similar patterns between drinking water arsenic and sweat chloride concentration. We were unable to assess how differences in the efficiency of arsenic metabolism between individuals, as might be demonstrated by relative distribution of urine arsenic species, affects associations between arsenic and sweat chloride concentration.

Our studies translate studies from experimental models [20,22] into humans and suggest that one mechanism of arsenic toxicity might include inducing the degradation of CFTR. In these models, CFTR dysfunction is associated with decreased of surface liquid volume, similar to what is seen in the genetic disease cystic fibrosis, where the absence or dysfunction of CFTR in airway epithelium leads to thickened secretions and reduced mucociliary transport, resulting in mucus retention and plugging of airways, favoring persistent bacterial infections and inflammation [39].

**Table 4. Toenail arsenic (µg/g) and measures of lung function. $FEV_1$, FVC, and $FEV_1$/FVC are modeled using multivariable linear regression. Airflow obstruction is modeled using logistic regression. β represents change per IQR toenail arsenic concentration.**

| Toenail arsenic | $FEV_1$ | | FVC | | $FEV_1$/FVC | | Airflow obstruction | |
|---|---|---|---|---|---|---|---|---|
| | β (95% CI) † | *P*-value | β (95% CI) † | *P*-value | β (95% CI) † | *P*-value | OR (95% CI) † | *P*-value |
| 2001-2003 | −0.05 (−0.11, 0.02) | 0.15 | −0.03 (−0.10, 0.03) | 0.30 | −0.01 (−0.02, 0.002) | 0.11 | 1.57 (1.03, 2.40) | 0.04 |
| 2018-2021 | −0.05 (−0.13, 0.03) | 0.23 | −0.03 (−0.11, 0.06) | 0.54 | −0.01 (−0.03, 0.005) | 0.17 | 1.41 (0.82, 2.45) | 0.22 |
| Long-term‡ | −0.08 (−0.16, 0.01) | 0.09 | −0.05 (−0.14, 0.04) | 0.29 | −0.02 (−0.03, 0.001) | 0.06 | 1.97 (1.06, 3.67) | 0.03 |

†Models adjusted for age, sex, smoking status (current, former or never), education (able to write, primary education, middle school or above), and height.

‡Average of toenail arsenic concentrations in 2001–2003 and 2018–2021.

Our findings showing associations between arsenic and airway obstruction are consistent with previous studies that describe a relationship between arsenic exposure and lung dysfunction [3,5–8]. Cross-sectional and cohort studies among childhood and adult populations in China, Mexico, and the United States consistently demonstrate relationships between arsenic exposure and restrictive patterns on lung function testing that are suggestive of restrictive lung disease [40–43]. Indeed, in a meta-analysis of nine studies totaling 4,699 participants in Bangladesh, Chile, India, Mexico, and Pakistan, higher arsenic exposure as measured by urine or drinking water concentrations was associated with reduced $FEV_1$ and FVC, but not $FEV_1$/FVC [44]. In contrast, we found a relationship between arsenic and airflow obstruction (reduced $FEV_1$/FVC), and statistically insignificant reductions in $FEV_1$ and FVC. Relying upon spirometry to diagnose restrictive lung disease must be done with caution: total lung capacity (TLC) is the diagnostic gold standard to diagnose restrictive lung disease, and although FVC and TLC are related lung measurements, FVC-based algorithms unreliably identify restrictive lung disease [45]. Importantly, the odds of airflow obstruction among those with higher arsenic exposure were more pronounced among participants in our study who had never smoked, which suggests that non-smoking populations are particularly vulnerable to the lung toxicity of long-term arsenic exposure.

The magnitude of observed association between toenail arsenic and airflow obstruction is considerable when compared with the estimated effect of other covariates. For example, the estimated effect of toenail arsenic on airflow obstruction [β = 0.677 (OR 1.97, 95% CI: 1.057, 3.670) per IQR] is comparable to, and even greater than, the estimated effect of a 10-year increase in age [β = 0.604 (OR 1.83, 95% CI: 1.05, 3.18) per 10 years] in regression model adjusted for covariates (S5 Table), suggesting that arsenic exposure has a significant impact on airflow obstruction.

We did not find evidence that sweat chloride mediates the relationship between arsenic exposure and airflow obstruction, which suggests that there are mechanisms other than CFTR dysfunction that may explain arsenic's associations with lung health. For example, arsenic has been shown to induce direct lung tissue injury in tracheal epithelial cells and alveolar macrophages through the generation of reactive oxygen species and stress proteins [46,47]. Arsenic exposure is also associated with decreased serum CC16 levels [48], which is a lung protective protein secreted by alveolar epithelial cells that modulates inflammatory lung injury. Lastly, arsenic exposure is associated with increased matrix metalloproteinase-9, one of a class of proteins that regulate airway inflammation and have been implicated in the pathogenesis of obstructive lung disease [49–51].

Our study has multiple implications for screening of arsenic-related disease in countries such as Bangladesh. Sweat tests may be an effective, inexpensive, point-of-care screening test for arsenic exposure. As we have demonstrated here and previously [24], it is possible to perform sweat tests in rural areas without expensive laboratory equipment, although the costs of commercially available instruments and consumables remain high for widespread implementation in low and middle income countries (LMICs). Another implication of our finding is that diagnoses of cystic fibrosis made only by sweat tests (i.e., without genetic testing, as is done in LMICs including Bangladesh [52]) may be false positives – that is, individuals' elevated sweat chloride concentrations may be a result of high arsenic exposure and not genetic disease. In countries with high arsenic exposure, when cystic fibrosis is suspected because of a constellation of symptoms such as lung disease, fat malabsorption, and diabetes, evaluating for arsenic exposure might complement sweat tests, and genetic confirmation of cystic fibrosis may be warranted.

In a similar manner, better understanding of the mechanisms of arsenic toxicity may benefit individuals with genetically-determined cystic fibrosis. Not all the individuals in our study who were exposed to high concentrations of arsenic had elevated sweat chloride concentrations or lung dysfunction, suggesting that other factors may be associated with preserved epithelial chloride transport. These factors might reveal protective mechanisms that could be exploited to develop therapies for genetically determined cystic fibrosis. The recent work in killifish and cell culture described above [20–22] suggests such mechanisms might include suppressed cellular response of ubiquitinylation.

The strengths of our study include our direct measurements of individual-level long-term arsenic exposure using toenail clippings, sweat chloride concentrations, lung function, and – among those with elevated sweat chloride concentrations

– genotyping to exclude pathogenic *CFTR* variants. None of the samples tested revealed a genetic diagnosis of cystic fibrosis, that is, no participant had homozygous or compound heterozygous mutations in *CFTR* in variants known to be associated with a cystic fibrosis phenotype. Our study also has important limitations. First, as with all cross-sectional studies, because sweat chloride concentrations were measured at the same time that water and toenail samples were collected, we cannot confirm that arsenic exposure preceded their elevated sweat chloride concentrations. However, the prevalence of *CFTR* mutations in South Asian populations is low [53]. This, in combination with the bench literature supporting CFTR dysfunction as a plausible mechanism through which arsenic exposure causes high sweat chloride concentrations, we believe our observations likely represent a true association with arsenic exposure. Future longitudinal studies are needed to establish a causal relationship between arsenic exposure, sweat chloride concentrations, and lung function. We also obtained PFTs in only a subset of the total study population because the COVID-19 pandemic interrupted PFT testing. However, there were no substantive differences in the participants who completed PFTs as compared to those who did not (S1 Table), so we do not anticipate that this biased our results.

We were only able to perform *CFTR* sequencing on 45 of the 69 participants with sweat chloride ≥ 60 mmol/L because of laboratory closures during the COVID-19 pandemic. Given the low rates of pathogenetic *CFTR* mutations in Bangladeshi populations and our sequencing results from the large subset, we do not anticipate that we missed ascertaining genetic diagnoses of cystic fibrosis among the individuals who did not undergo sequencing. We were not able to evaluate the role of heterozygous *CFTR* mutations on sweat chloride concentrations or arsenic-variant interactions, and this is an additional limitation of our study. Finally, assessment of CFTR function on the skin may not be wholly representative of activity in other tissues or cell types.

Air pollution is an important contributor to lung disease in Bangladesh [54,55]. Unfortunately, we do not have direct or surrogate measures of air quality in our study, so we are unable to adjust these in our analyses, and this is another important limitation of our study. Arsenic is also present in fine particulate matter ($PM_{2.5}$), much of it in potentially bioavailable forms [56,57], thus air pollution may also be an important source of arsenic exposure for some populations. Studies from China, for example, found high concentrations of arsenic in ambient $PM_{2.5}$ samples [56–58], and studies in rats using ambient $PM_{2.5}$ samples from Tangshan, China suggest that arsenic accumulates in blood and lung tissue following repeated respiratory exposure [58]. In Bangladesh, however, reports of arsenic concentrations in $PM_{2.5}$ are more than 10 times lower than those found in China [59], suggesting that the high levels of arsenic in contaminated drinking water is the main source of arsenic exposure for our study population. Future work will be needed to characterize the role of air pollution in the relationships between arsenic, sweat chloride, and lung function. Additionally, although we were able to adjust for sex, education, and height, there also may be residual confounding from unmeasured confounders such as nutritional status and occupation.

We were unable to characterize other pulmonary features of cystic fibrosis such as bronchiectasis, a finding that would be most evident on computed tomography (CT) imaging as compared to plain films, or non-pulmonary features of cystic fibrosis such as pancreatic dysfunction. Future work directly measuring CFTR function in tissues including lung epithelium, neutrophils, and pancreatic islet cells may be provide more insight into how arsenic exposure may affect CFTR function and potentially cause disease. Finally, our study population is from a specific region in Bangladesh with exposures to high arsenic concentrations, which may limit the generalizability of our findings to other countries. However, the arsenic measurements at the most recent assessment were within the range of those seen globally, which suggests that even moderately elevated arsenic concentrations may affect CFTR function.

## Conclusions

Environmental arsenic exposure is associated with impaired CFTR function, as demonstrated by elevated sweat chloride concentrations, and airway obstruction. Potential applications of our study include future work aimed at development of a point-of-care sweat chloride test as a biomarker of arsenic exposure.

## Supporting information

**S1 Table. Comparison of study population characteristics among those included in the PFT analysis compared to those for whom PFTs were not available.**
(DOCX)

**S2 Table. Relationship between sweat chloride and age, sex, education and smoking status.**
(DOCX)

**S3 Table. *CFTR* sequencing results.**
(DOCX)

**S4 Table. Adjusted effect estimates (β and 95% CIs) [†] from multivariable regression models for lung function and odds ratios (ORs and 95% CIs) [†] from logistic regression models for airflow obstruction associated with an IQR increase in toenail arsenic among never smokers (*n* = 115).**
(DOCX)

**S5 Table. Association between airflow obstruction and age, sex, height, and smoking status.**
(DOCX)

**S1 Fig. Distribution of long-term drinking water and toenail arsenic concentrations among participants with normal/intermediate (<60 mmol/L) and abnormal (≥60 mmol/L) sweat chloride concentration.**
(DOCX)

**S2 Fig. Odds ratios (OR) and 95% confidence intervals (CI) for abnormal sweat chloride at different cutoff levels associated with concurrent toenail arsenic concentration.** All models are adjusted for sex and smoking status.
(DOCX)

**S3 Fig. Odds ratios (OR) and 95% confidence intervals (CI) for abnormal sweat chloride at different cutoff levels associated with long-term toenail arsenic concentration.** All models are adjusted for sex and smoking status.
(DOCX)

**S4 Fig. Mediation analysis of the long-term arsenic-lung function by sweat chloride.** (A) Mediation for $FEV_1$; (B) Mediation for FVC; (C) Mediation for $FEV_1$/FVC; (D) Mediation for airway obstruction. Models were adjusted for age, sex, height, smoking status, and education.
(DOCX)

## Author contributions

**Conceptualization:** Subrata K. Biswas, Abby F. Fleisch, David S. Ludwig, Christopher Hug, Quazi Quamruzzaman, David C. Christiani, Maitreyi Mazumdar.

**Data curation:** Subrata K. Biswas, Afifah Farooque, Diane Bao.

**Formal analysis:** Mi-Sun S. Lee, Crystal M. North, Afifah Farooque.

**Funding acquisition:** Quazi Quamruzzaman, David C. Christiani, Maitreyi Mazumdar.

**Investigation:** Crystal M. North, Irada Choudhuri, Subrata K. Biswas, Sakila Afroz, Sadia Mow, Nazmul Husain, Fuadul Islam, Md Mostofa Golam, Partha Protim Biswas, Subba R. Digumarthy, Maitreyi Mazumdar.

**Methodology:** Crystal M. North, Subrata K. Biswas, Maitreyi Mazumdar.

**Project administration:** Diane Bao, Sakila Afroz, Sadia Mow, Md Mostofa Golam, Maitreyi Mazumdar.

**Resources:** David C. Christiani.

**Supervision:** Md Mostofa Golam, Quazi Quamruzzaman, Maitreyi Mazumdar.

**Writing – original draft:** Mi-Sun S. Lee, Crystal M. North.

**Writing – review & editing:** Mi-Sun S. Lee, Crystal M. North, Irada Choudhuri, Subrata K. Biswas, Abby F. Fleisch, Afifah Farooque, David S. Ludwig, Christopher Hug, Quazi Quamruzzaman, David C. Christiani, Maitreyi Mazumdar.

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
