## [Decision Letter · Decision Letter 0]

8 Dec 2024

PONE-D-24-41161Arsenic exposure is associated with elevated sweat chloride concentration and airflow obstruction among adults in Bangladesh: a cross sectional studyPLOS ONE

Dear Dr. Mazumdar,

Thank you for submitting your manuscript to PLOS ONE. After careful consideration, we feel that it has merit but does not fully meet PLOS ONE’s publication criteria as it currently stands. Therefore, we invite you to submit a revised version of the manuscript that addresses the points raised during the review process.

We look forward to receiving your revised manuscript.

Kind regards,

Aaron Specht

Academic Editor

PLOS ONE

Journal Requirements:

“We received funding from the National Institute of Environmental Health Sciences (R01 ES027825, R01 ES011622, P30 ES000002) and the National Heart, Lung, and Blood Institute (K23 HL154863).  “

“We received funding from the National Institute of Environmental Health Sciences (niehs.nih.gov) - R01 ES027825 (Mazumdar), R01 ES011622 (Christiani), P30 ES000002 (Mazumdar) and the National Heart, Lung, and Blood Institute (nhlbi.nih.gov) K23 HL154863(North). “

5. Please include a caption for figure 1.

6 Please include your tables as part of your main manuscript and remove the individual files. Please note that supplementary tables (should remain/ be uploaded) as separate "supporting information" files.

Reviewers' comments:

Reviewer's Responses to Questions

**Comments to the Author**

1. Is the manuscript technically sound, and do the data support the conclusions?

Reviewer #1: Partly

Reviewer #2: No

Reviewer #3: Yes

2. Has the statistical analysis been performed appropriately and rigorously? 

Reviewer #1: No

Reviewer #2: N/A

Reviewer #3: Yes

3. Have the authors made all data underlying the findings in their manuscript fully available?

Reviewer #1: Yes

Reviewer #2: No

Reviewer #3: Yes

4. Is the manuscript presented in an intelligible fashion and written in standard English?

Reviewer #1: Yes

Reviewer #2: Yes

Reviewer #3: Yes

5. Review Comments to the Author

Reviewer #1: Major Comment:

This study investigates the relationship between arsenic exposure, sweat chloride levels (a marker of cystic fibrosis transmembrane conductance regulator [CFTR] function), and pulmonary function among adults in Bangladesh. The study presents intriguing findings suggesting that elevated sweat chloride may be a novel biomarker for arsenic exposure while raising questions about the role of CFTR dysfunction in arsenic-induced lung disease. The study is generally well-conducted, with a clear focus, appropriate methodology, and comprehensive data analysis. However, there are areas where the manuscript could be improved in terms of clarity, precision, and discussion of the limitations and implications of the findings.

Advantage:

1) The study addresses a significant public health issue: arsenic exposure and its potential impact on lung health in a highly exposed population. However, PM2.5 and other co-morbidities with air pollutants should be adjusted and discussed as confounding factors.

2) The study design, including the recruitment of participants, collection of biological samples, and measurement of variables, is appropriate for addressing the research question. The multiple exposures should reveal the relationship and interaction between lung function and CFTR in the design of further studies.

3) The statistical analysis is comprehensive and appropriate for the data collected. The authors have explored the relationships between the variables of interest using various statistical methods, including linear regression, logistic regression, and mediation analysis. The internal and external doses of arsenic should be discussed comprehensively, and distribution should fit the assumption of normal distribution. Thus, the un-transformation and transformation of indicators should be listed in tables, and the non-parametric methods should even be chosen.

4) The study provides novel evidence suggesting that elevated sweat chloride may be a potential biomarker for arsenic exposure. This finding has important implications for the screening and diagnosing arsenic-related health issues. The concordance of arsenic exposure between sweat chloride and blood or urine arsenic metabolites, such as DMA and MMA levels, should be discussed.

5) The manuscript is generally well-written and organized, making it easy to follow the study's rationale, methods, results, and conclusions; however, the correlation between sweat chloride and arsenic exposure should also identify the time-lapse changes.

6) The study population is drawn from a specific region in Bangladesh with high arsenic exposure. The authors need to explicitly acknowledge that the findings may need to be more generalizable to other populations with different levels of arsenic exposure or genetic backgrounds.

7) The study's cross-sectional nature limits the ability to draw definitive conclusions about causality. The authors must emphasize this limitation and suggest future longitudinal studies to establish a temporal relationship between arsenic exposure, sweat chloride levels, and lung function.

8) The manuscript needs more detailed information on the types of lung function tests conducted and the criteria used to define airflow obstruction. More specific information about the lung function assessment would strengthen the study's validity.

9) The study adjusts for several potential confounders. Still, other unmeasured factors, such as nutritional status, socioeconomic status, and exposure to different environmental pollutants, may influence the relationship between arsenic and lung function. The authors should address this limitation.

10) While the study did not find evidence of CFTR mutations in participants with elevated sweat chloride, it does not fully explore other potential mechanisms of CFTR dysfunction that may be induced by arsenic exposure. Expanding the discussion on this aspect would enhance the study's contribution to understanding arsenic toxicity.

Minor Comments:

1) Introduction: The introduction provides a good overview of the problem and the rationale for the study. However, it could be strengthened by briefly mentioning the specific aims and hypotheses of the study.

2) Methods: The methods section is generally well-described. However, providing more details on the quality control measures used for the sweat chloride and lung function testing would be helpful.

3) Results: The results are presented clearly and concisely. However, more information on the distribution of sweat chloride levels in the study population, such as the interquartile range or the 95% confidence interval, would be helpful.

4) Discussion: The discussion reasonably interprets the findings and their implications. However, it could be improved by discussing the study's limitations more explicitly and suggesting future research directions.

This study provides impressive insights into the relationship between arsenic exposure, sweat chloride levels, and lung function. The findings have important implications for public health, particularly in regions with high arsenic contamination. By addressing the negative comments and incorporating specific suggestions, the authors can significantly enhance their manuscript's clarity, precision, and impact.

Reviewer #2: 

- Please provide the IRB approval information of this study.

- In Figure 1, I may assume the comparison did not consider the impact of other factors on levels of sweat chloride. If this is true, then how to convince the readers the difference was mainly driven by arsenic exposure?

- Please verify the linearity assumption for association between As exposure and levels of sweat chloride.

- In mediation analysis, please specify what covariates were adjusted for mediator and outcome models. Usually the confounding factors in these two models should be different.

-

Reviewer #3: The manuscript titled "Arsenic exposure is associated with elevated sweat chloride concentration and airflow obstruction among adults in Bangladesh: a cross-sectional study" investigates the link between arsenic exposure, CFTR dysfunction (as measured by sweat chloride levels), and pulmonary function in Bangladeshi adults. The authors hypothesize that arsenic exposure impairs CFTR function, reflected in elevated sweat chloride concentrations, and that this dysfunction may mediate arsenic’s impact on lung function. Through comprehensive measurements of arsenic levels in toenails and water, sweat chloride testing, pulmonary function testing, and CFTR genotyping, the study aims to establish sweat chloride as a biomarker for arsenic exposure and explore the mechanisms underlying arsenic-associated lung disease. This study is significant due to its focus on an underexplored potential biomarker (sweat chloride) for arsenic exposure and its implications for screening arsenic-related diseases in low-resource settings. However, the manuscript should only be considered after proper adjustments and corrections have been made including clarity in methodology, coherence in data interpretation, and justification of certain scientific claims. Kindly see attachement for all comments.

6. PLOS authors have the option to publish the peer review history of their article (what does this mean? ). If published, this will include your full peer review and any attached files.

**Do you want your identity to be public for this peer review?** For information about this choice, including consent withdrawal, please see our Privacy Policy .

Reviewer #1: No

Reviewer #2: No

Reviewer #3: No

---

## [Author Response · Author response to Decision Letter 1]

3 Feb 2025

29 January 2025

Dr. Aaron Specht

Academic Editor

PLOS One

RE: PONE-D_24-41161 REVISION “Arsenic exposure is associated with elevated sweat chloride concentration and airflow obstruction among adults in Bangladesh: a cross-sectional study.”

Dear Dr. Specht,

I am writing to submit a revised version of our manuscript titled, “Arsenic exposure is associated with elevated sweat chloride concentration and airflow obstruction among adults in Bangladesh: a cross-sectional study.” We greatly appreciate the constructive feedback provided by the reviewers and the opportunity to revise our work considering their comments.

In response to the reviewers’ suggestions, we have made several changes to improve manuscript. Below is a point-by-point reponse to the issues raised in the review.

1. Please ensure that your manuscript meets PLOSOne’s style requirements, including those for file naming.

Response: We have reformatted our manuscript to be consistent with PLOSOne’s style requirements.

2. Please include a complete copy of PLOSOne’s questionnaire on inclusivity in global research.

Response: We have included a complete copy of PLOSOne’s questionnaire on inclusivity in global research and have uploaded it as Supporting Information.

3. Data availability on request

Response: We have uploaded a minimal anonymized dataset to the Harvard Dataverse. The dataset can cam be found at the following location: https://dataverse.harvard.edu/dataset.xhtml?persistentId=doi:10.7910/DVN/OLY5P2

4. Please remove any funding-related text from the manuscript and let us know how you would like to update your funding statement.

Response: We have removed funding information from the manuscript. Our funding statement should read as follows: “We received funding from the National Institute of Environmental Health Sciences (niehs.nih.gov) - R01 ES027825 (Mazumdar), R01 ES011622 (Christiani), P30 ES000002 (Mazumdar) and the National Heart, Lung, and Blood Institute (nhlbi.nih.gov) K23 HL154863(North). The contents are solely the responsibility of the authors and do not represent the official views of the National Institutes of Health.“

5. Please include a caption for Figure 1.

Response: We have added a caption for Figure 1, p. 17, line 438

6. Please include your tables as part of your main manuscript and remove the individual files. Please note that supplementary tables (should remain/ be uploaded) as separate "supporting information" files.

Response: We have included the tables in the manuscript and the supplementary tables remain in a separate file.

7. Please include captions for your Supporting Information files at the end of your manuscript, and update any in-text citations to match accordingly.

Response: We have updated the captions and in-text citations for supporting information as directed.

Reviewer #1:

1) The study addresses a significant public health issue: arsenic exposure and its potential impact on lung health in a highly exposed population. However, PM2.5 and other co-morbidities with air pollutants should be adjusted and discussed as confounding factors.

Response: We expand our discussion of the this limitation. Below is a key paragraph from this discussion that addresses the concerns of the reviewer.

p. 24, line 472. Air pollution is an important contributor to lung disease in Bangladesh[1, 2]. We did not measure air quality, thus are unable to adjust for PM2.5 exposure and other components of air pollution. However, we do not have reason to suspect that those exposed to higher as compared to lower arsenic levels would have differential air pollution exposure, and would therefore expect that not adjusting for air pollution exposure would bias our results towards the null, further emphasizing the importance of our findings. That we found associations between arsenic and airflow exposure despite the absence of air pollution data suggests that the relationship between arsenic and lung function might be stronger than what is observed in our study. In fact, the presence of airflow obstruction among the largely non-smoking women in our cohort further underscores the importance of air pollution as a cause of obstructive lung disease in the region. Additionally, although we were able to adjust for sex, education, and height, there also may be residual confounding from unmeasured confounders such as nutritional status and occupation.

2) The study design, including the recruitment of participants, collection of biological samples, and measurement of variables, is appropriate for addressing the research question. The multiple exposures should reveal the relationship and interaction between lung function and CFTR in the design of further studies.

Response: We have not made any changes in response to this comment.

3) The statistical analysis is comprehensive and appropriate for the data collected. The authors have explored the relationships between the variables of interest using various statistical methods, including linear regression, logistic regression, and mediation analysis. The internal and external doses of arsenic should be discussed comprehensively, and distribution should fit the assumption of normal distribution. Thus, the un-transformation and transformation of indicators should be listed in tables, and the non-parametric methods should even be chosen.

Response: Thank you for the insightful comments. In many observational studies, biomarker data often do not follow a normal distribution. However, linear regression still can be used even when the data are not perfectly normal. Non-parametric methods typically have lower power to detect true effects and often require larger sample sizes (Li et al. 2012). They generally provide less information about the data as they typically test for differences in ranks rather than actual values, potentially limiting the depth of information about the data. In our study, we examined a comprehensive visual inspection using histograms and kernel density plots, along with residual distribution analysis (see Figure below). These analyses suggested that the data appeared approximately normal, with no major outliers or pattern that might affect the model’s validity. Using untransformed values in regression analysis allow the coefficients to represent the effect of a one-unit change in the predictor on the outcome, which can make the results easier to interpret. According to the Central Limit Theorem (CLT) with the adequacy of sample size (Li et al. 2012), the estimated regression coefficients remain reliable. In addition, a simulation study even with heavily skewed data, the cofficients, study power, and confidence interval were fairly reliable at least with a sample size of 100 or larger (Knief and Forstmeier 2021). Based on these considerations, we chose to use a parametric approach in our analysis.

References (added to the manuscript): Li X, Wong W, Lamoureux EL, Wong TY. Are linear regression techniques appropriate for analysis when the dependent (outcome) variable is not normally distributed? Invest Ophthalmol Vis Sci. 2012;53(6):3082-3. doi: 10.1167/iovs.12-9967. PMID: 22618757.

Knief U, Forstmeier W. Violating the normality assumption may be the lesser of two evils. Behav Res Methods. 2021;53(6):2576-2590. doi: 10.3758/s13428-021-01587-5. PMID: 33963496.

Figure: (Please see attached cover letter) Histogram and kernel density of sweat chloride concentration superimposed. The blue solid line illustrates the normal distribution; the red solid line illustrates the kernel distribution; the blue dashed line presents median of sweat chloride (48 mmol/L). (right) Residual distribution

We have made the following additions to our manuscript to clarify these points.

1). We add comments and citations about the use of toenails to estimate internal doses of arsenic as below:

p. 19, line 371. We used toenail arsenic concentrations to estimate internal arsenic dose. Toenail arsenic is considered good biomarker for long-term exposure when external exposures are consistent[3]. In secondary analyses, despite the modest correlation between toenail and drinking water arsenic, we found similar patterns between drinking water arsenic and sweat chloride concentration. We were unable to assess how differences in the efficiency of arsenic metabolism between individuals, as demonstrated by relative distribution of urine arsenic species, affects associations between arsenic and sweat chloride concentration.

2) We summarize our tests of normality and implications for choice of modeling as below:

p.16, line 310. Our decisions to use untransformed arsenic values and parametric models are supported by a comprehensive visual inspection using histograms and kernel density plots, along with residual distribution analysis (data not shown), suggested that the data appeared approximately normally distributed and the adequacy of our study’s sample size [36, 37].

4) The study provides novel evidence suggesting that elevated sweat chloride may be a potential biomarker for arsenic exposure. This finding has important implications for the screening and diagnosing arsenic-related health issues. The concordance of arsenic exposure between sweat chloride and blood or urine arsenic metabolites, such as DMA and MMA levels, should be discussed.

Response: We do not have measures of arsenic metabolites and have added this to the discussion below:

p. 20, line 375. We were unable to assess how differences in the efficiency of arsenic metabolism between individuals, as demonstrated by relative distribution of urine arsenic species, affects associations between arsenic and sweat chloride concentration.

5) The manuscript is generally well-written and organized, making it easy to follow the study's rationale, methods, results, and conclusions; however, the correlation between sweat chloride and arsenic exposure should also identify the time-lapse changes.

Response: We clarify that our primary analysis uses drinking water, toenails and sweat chloride that were measured on the same day.

p.16, line 314. In our primary analyses, we found that concurrent drinking water and toenail arsenic concentrations were higher among participants with sweat chloride ≥ 60 mmol/L compared to < 60 mmol/L

p.23, line 451. First, as with all cross-sectional studies, because sweat chloride concentrations were measured at the same time that water and toenail samples were collected, we cannot confirm that arsenic exposure preceded their elevated sweat chloride concentrations.

Secondary analyses assess measures of chronic arsenic exposure and are now placed in the Supporting material.

6) The study population is drawn from a specific region in Bangladesh with high arsenic exposure. The authors need to explicitly acknowledge that the findings may need to be more generalizable to other populations with different levels of arsenic exposure or genetic backgrounds.

Response: We added a statement to our limitation section acknowledging that our population is drawn from a specific region of Bangladesh with high arsenic exposure and this may limit generalizability.

p. 25, line 491. Finally, our study population is from a specific region in Bangladesh with exposures to high arsenic concentrations, which may limit the generalizability of our findings to other countries. However, the arsenic measurements at the most recent assessment were within the range of those seen globally, which suggests that even moderately elevated arsenic concentrations may affect CFTR function.

7) The study's cross-sectional nature limits the ability to draw definitive conclusions about causality. The authors must emphasize this limitation and suggest future longitudinal studies to establish a temporal relationship between arsenic exposure, sweat chloride levels, and lung function.

Response: We added a text to emphasize the cross-sectional nature of the study and need for longitudinal studies on p. 23, line 451.

First, as with all cross-sectional studies, because sweat chloride concentrations were measured at the same time that water and toenail samples were collected, we cannot confirm that arsenic exposure preceded their elevated sweat chloride concentrations.

8) The manuscript needs more detailed information on the types of lung function tests conducted and the criteria used to define airflow obstruction. More specific information about the lung function assessment would strengthen the study's validity.

Response: We provide more information about lung function tests on page 9, line 166, including the criteria used to define airflow obstruction. We follow (and provide citations for) the American Thoracic Society (ATS) guidelines for spirometry and the Global Initative for Chronic Obstructive Lung Disease (GOLD)’s criteria for airflow obstruction.

We measured lung function using EasyOne® Air handheld spirometers (Medical Technologies, Andover, MA). All participants had abstained from smoking for at least 45 minutes prior to testing and completed testing in the seated position after loosening any tight-fitting clothing. Spirometry was conducted in accordance with American Thoracic Society (ATS) guidelines by trained study staff, and participants were given up to seven tries to achieve three ATS acceptable and reproducible maneuvers[6]. Measured parameters included forced expiratory volume in one second (FEV1) and forced vital capacity (FVC), and all participants repeated PFTs after receiving four puffs of salbutamol (Beximco Pharmaceuticals, Bangladesh) and waiting for 10 minutes. Study staff monitored data quality control per ATS guidelines by assessing the accuracy of spirometer measurements with volume and multi-flow calibration using a 3-litre syringe (ndd Medical Technologies, Andover, MA) each day before study procedures began to ensure that measured flow and volumes remained within ± 3.5%, per ATS guidelines[6].

9) The study adjusts for several potential confounders. Still, other unmeasured factors, such as nutritional status, socioeconomic status, and exposure to different environmental pollutants, may influence the relationship between arsenic and lung function. The authors should address this limitation.

Response: We have added text to our discussion about this limitation.

p. 24, line 483. Although we were able to adjust for sex, education, and height, there also may be residual confounding from unmeasured confounders such as nutritional status and occupation.

10) While the study did not find evidence of CFTR mutations in participants with elevated sweat chloride, it does not fully explore other potential mechanisms of CFTR dysfunction that may be induced by arsenic exposure. Expanding the discussion on this aspect would enhance the study's contribution to understanding arsenic toxicity.

Response: We have made our hypothesis that arsenic degrades the CFTR protein after translation more explicit in our introduction. We do not suggest that arsenic causes mutations in CFTR.

p. 5 line 78. Our study aims to further translate these findings to humans and test the hypothesis that arsenic induces CFTR degradation in humans who are exposed to arsenic through the environment.

p. 5 line 95. Our study aimed to test the specific hypothesis that higher arsenic exposures lead to degradation of the CFTR protein in cell membranes; evidence of CFTR degradation is elevated sweat chloride.

Minor Comments:

1) Introduction: The introduction provides a good overview of the problem and the rationale for the study. However, it could be strengthened by briefly mentioning the specific aims and hypotheses of the study.

Response: We have made our hypothesis that arsenic degrades the CFTR protein more explicit in our introduction.

p. 5 line 78. Our study aims to further translate these findings to humans and test the hypothesis that arsenic induces CFTR degradation in humans who are exposed to arsenic through the environment.

p. 5 line 95. Our study aimed to test the specific hypothesis that higher arsenic exposures lead to degradation of the CFTR protein in cell membranes; evidence of CFTR degradation is elevated sweat chloride.

2) Methods: The methods s

---

## [Decision Letter · Decision Letter 1]

17 Mar 2025

Arsenic exposure is associated with elevated sweat chloride concentration and airflow obstruction among adults in Bangladesh: a cross sectional study

PONE-D-24-41161R1

Dear Dr. Mazumdar,

We’re pleased to inform you that your manuscript has been judged scientifically suitable for publication and will be formally accepted for publication once it meets all outstanding technical requirements.

Kind regards,

Aaron Specht

Academic Editor

PLOS ONE

Additional Editor Comments (optional):

Reviewers' comments:

Reviewer's Responses to Questions

**Comments to the Author**

1. If the authors have adequately addressed your comments raised in a previous round of review and you feel that this manuscript is now acceptable for publication, you may indicate that here to bypass the “Comments to the Author” section, enter your conflict of interest statement in the “Confidential to Editor” section, and submit your "Accept" recommendation.

Reviewer #1: (No Response)

Reviewer #3: All comments have been addressed

2. Is the manuscript technically sound, and do the data support the conclusions?

Reviewer #1: Partly

Reviewer #3: Yes

3. Has the statistical analysis been performed appropriately and rigorously? 

Reviewer #1: Yes

Reviewer #3: Yes

4. Have the authors made all data underlying the findings in their manuscript fully available?

Reviewer #1: Yes

Reviewer #3: Yes

5. Is the manuscript presented in an intelligible fashion and written in standard English?

Reviewer #1: Yes

Reviewer #3: Yes

6. Review Comments to the Author

Reviewer #1: The authors would not discuss the PM2.5 and arsenic exposure interaction, and the comprehensive view was not standing on the solid evidence, so the previous study demonstrated the PM2.5 existing arsenic fraction and represented elevation of risk (https://doi.org/10.1016/j.envpol.2019.113881), which authors should not omitted; thus, the biases should not null and need to discuss the PM2.5 and arsenic exposure interaction, even the cumulative external dose might be played as fallacy for assessment risk for arsenic exposure, which represented why authors need to convince us on scientific view. From a comprehensive point of view, the overall evidence might not stand on solid ground; authors need to discuss this in as much detail as possible. So the previous study demonstrated that the PM2.5 existing arsenic fraction and represented elevation of risk was not a coincidence; thus, the biases should not be null. I encourage these authors to find surrogate markers to adjust PM2.5 levels and re-analyse the risk of external arsenic exposure. Even with no more evidence, they should be humble enough to face the weakness in the present study.

Reviewer #3: The in-text citations should be formatted consistently throughout the manuscript. Currently, there are inconsistencies, such as in line 120 and line 420, and throughout the manuscript, the citations are placed directly after the last word without a space. To ensure clarity and adherence to proper citation formatting, the authors should include a single space between the last word and the citation number in all instances.

7. PLOS authors have the option to publish the peer review history of their article (what does this mean? ). If published, this will include your full peer review and any attached files.

**Do you want your identity to be public for this peer review?** For information about this choice, including consent withdrawal, please see our Privacy Policy .

Reviewer #1: No

Reviewer #3: No

---

## [Editor Report · Acceptance letter]

PONE-D-24-41161R1

PLOS ONE

Dear Dr. Mazumdar,

I'm pleased to inform you that your manuscript has been deemed suitable for publication in PLOS ONE. Congratulations! Your manuscript is now being handed over to our production team.

Kind regards,

on behalf of

Dr. Aaron Specht

Academic Editor

PLOS ONE